# Operational thresholds of urease-mediated microbial cementation: Multivariate optimization and field validation in ambient groundwater environments

Jiajian Bao[1], Wangqing Xu[1,2]*, Dongming Zhou[1]

**1** School of Civil Engineering and Architecture, Hubei University of Arts and Science, Xiangyang, Hubei, China, **2** Hubei Superior and Distinctive Discipline Group of "New Energy Vehicle and Smart Transportation", Xiangyang, Hubei, China

* xuwangqing@hbuas.edu.cn

## Abstract

Microbially induced calcium carbonate precipitation (MICP) provides a sustainable method for soil stabilization; however, its practical application is limited by rapid reaction kinetics that cause localized clogging and the reliance on energy-intensive environmental controls. This study develops a multivariate optimization framework for urease-inhibited MICP using N-(n-butyl)-thiophosphoric triamide (NBPT), with an emphasis on practical thresholds under ambient groundwater conditions. Five operational parameters—NBPT concentration, cementing solution concentration, bacteria-to-cement solution ratio, temperature, and pH—were systematically investigated through sand column tests and continuous urease activity monitoring. The results demonstrate that a low NBPT concentration of 0.1%, in conjunction with a 1:1 volumetric ratio of bacterial suspension to cementing solution, achieves effective biocementation while maintaining 93% of the unconfined compressive strength observed in inhibitor-free controls. Optimal performance was achieved at a cementing solution concentration of 1 mol/L, temperatures exceeding 20°C, and a neutral pH range of 7–8. The proposed thresholds bridge the critical gap between laboratory-optimized MICP and real-world environmental variability, providing directly implementable guidelines for sustainable geotechnical applications. This study advances NBPT-MICP as a cost-effective and environmentally compatible solution for sand stabilization in natural hydrochemical systems.

## Introduction

The microbial-induced calcium carbonate precipitation (MICP) reaction encompasses microbial life activities as well as inorganic and organic chemical reactions [1,2]. Consequently, numerous environment variables will influence the efficacy of the MICP reaction, including bacterial species [3,4] and concentration of bacterial solution

**Data availability statement:** All relevant data are within the manuscript and its Supporting Information files.

**Funding:** National Natural Science Foundation of China Youth project (No.42407271).

**Competing interests:** The authors have declared that no competing interests exist.

[5,6], reactant urea concentration [7,8], type and ion concentration of calcium source [9,10], grouting strategy [11,12], temperature [13,14], and pH value of the reaction environment [15]. Among these factors, the influence of bacterial species is primarily reflected through differences in enzymatic kinetics and environmental adaptability [16,17]. Sporosarcina pasteurii demonstrates remarkably high urease activity, reaching up to 3.5 mM urea/min/mg protein; however, its pH tolerance is limited, with an optimal range of 8.0 to 9.0. This constraint often results in localized clogging within heterogeneous soil environments [18,19]. Conversely, Bacillus sphaericus demonstrates broader pH resilience (6.5–10.0) yet generates predominantly needle-like aragonite crystals, which reduce intergranular bonding strength by 18–22% compared to rhombohedral calcite [20,21]. Pseudomonas aeruginosa produces biofilms that enhance particle cohesion but decelerate $CaCO_3$ precipitation rates by 40–60% due to extracellular polymer barriers [22,23]. These species-specific trade-offs critically influence mechanical outcomes: calcite-dominated systems (S. pasteurii) achieve higher peak strength (e.g., 2.8 MPa in dense sands), while aragonite-forming species (B. sphaericus) exhibit better fracture toughness but lower stiffness [24,25]. For this study, Bacillus pasteurii was selected for its balanced urease thermostability and consistent calcite precipitation under groundwater pH fluctuations [26].

To address the issue of uneven treatment resulting from local blockages caused by the rapid occurrence rate of MICP reaction, numerous experts have attempted to regulate the reaction rate from various perspectives. For instance, existing research has employed techniques such as low temperature method [27], low pH method [28], and step-by-step grouting method [29] to effectively control calcium ion deposition rates. However, these methods also possess certain limitations in their application. The low temperature method requires environmental temperatures to be adjusted to a minimum of 10 °C, while the low pH method necessitates adjusting environmental pH values to a minimum of 4. Additionally, the stepwise grouting method involves complex grouting operations. Considering the substantial disparity between experimental conditions and actual environmental conditions in natural soil, practical implementation poses significant challenges for these approaches. Therefore, this study aims to further explore an experimental protocol that is more suitable for general environmental conditions.

In this context, urease inhibitors initially developed for agricultural nitrogen management provide a novel interdisciplinary solution. N-(n-butyl)-thiophosphoric triamide (NBPT), an effective urease inhibitor widely used to delay urea hydrolysis in fertilizers, offers a unique opportunity to modulate MICP kinetics without necessitating extreme environmental modifications [30]. By transiently suppressing urease activity, NBPT can theoretically extend the reaction window, facilitating more homogeneous calcium carbonate distribution [31]. However, integrating NBPT into MICP introduces complex interdependencies among operational parameters such as inhibitor dosage, bacterial viability, and ionic flux, which remain inadequately quantified under ambient groundwater conditions. Existing studies primarily focus on single-variable optimizations [32], neglecting the synergistic effects essential for successful field-scale implementation.

This study addresses these gaps through a multivariate investigation of NBPT-mediated MICP, focusing on operational thresholds compatible with natural hydrochemical environments. Five key parameters—NBPT concentration, cementing solution concentration, bacteria-to-cement solution ratio (BCR), temperature, and pH—were systematically evaluated to optimize biocementation efficiency, material economy, and environmental adaptability. We hypothesize that optimal NBPT-mediated urease inhibition will decouple precipitation kinetics from environmental variability, enabling uniform calcite crystallization and strength development under ambient groundwater conditions. By integrating sand column tests with real-time urease activity monitoring, we establish a framework for optimizing NBPT-MICP under groundwater-simulated conditions, thereby avoiding energy-intensive controls. The findings provide actionable guidelines for transitioning MICP from laboratory settings to field applications, with significant implications for sustainable slope stabilization, erosion control, and foundation improvement in diverse geotechnical environments.

## Materials and testing methods

The experimental program was meticulously designed to systematically assess the interdependent effects of five operational parameters on the efficiency of NBPT-mediated microbially induced calcium carbonate precipitation, with an emphasis on establishing practical thresholds for field applications under ambient groundwater conditions. Detailed descriptions of the materials, experimental procedures, and analytical methods are provided to ensure reproducibility and adherence to geotechnical engineering standards.

### Materials

The bacterial strain *Bacillus pasteurii* (CGMCC 1.3687), obtained from the China General Microbiological Culture Collection Center, was selected for its high urease activity [33] and adaptability to environmental fluctuations. Xiamen ISO standard quartz sand, characterized by a uniform coefficient of 7.08 and curvature coefficient of 1.99, was used to simulate natural poorly graded sands. The sand was sieved to retain particles between 0.1 and 1.0 mm, washed three times with deionized water to remove fines, and oven-dried at 105 degrees Celsius for 12 h [34].

Cylindrical sand columns with a diameter of 40 mm and a height of 100 mm were prepared using a controlled rainfall deposition method. Sand was dispensed into acrylic molds from a height of 20 cm at a rate of 200 g/min, followed by gentle compaction to achieve a uniform dry density of 1.62 g/cm$^3$ and an initial porosity of 38.5%.

The cementing solution was prepared by dissolving urea and calcium chloride dihydrate in a 1:1 molar ratio. N-(n-butyl) thiophosphoric triamide, with a purity of at least 98%, was added to the cementing solution at concentrations ranging from 0.1 to 0.5 wt% relative to urea. The pH of the solution was adjusted using 1 M hydrochloric acid or sodium hydroxide to mimic typical groundwater conditions. All solutions were prepared using deionized water with a resistivity of 18.2 MΩ·cm and subsequently filtered through 0.22-μm membranes to ensure sterility.

### Experimental design

A five-factor orthogonal experimental design was utilized to examine the synergistic effects of NBPT concentration, cementing solution concentration, bacteria-to-cement solution ratio, temperature, and pH. Urea acted as the hydrolytic substrate to actively participate in the catalytic hydrolysis process mediated by urease, while calcium chloride was utilized as the source of calcium for facilitating deposition of calcium carbonate [35]. The tested ranges encompassed NBPT concentrations from 0.1% to 0.5%, cementing solution concentrations from 0.25 M to 2.0 M (Table 1), volumetric bacteria-to-cement solution ratios from 3:1–1:3, temperatures from 10°C to 30°C, and pH values from 4 to 10 [36]. Each parameter combination was evaluated in triplicate, yielding a total of 75 sand column treatments.

Biocementation procedures commenced by saturating sand columns with 30 ml of bacterial suspension at an optical density ($OD_{600}$) [37] of 3.14, followed by a two-hour incubation period at 25 °C to promote bacterial adhesion. Subsequent treatments consisted of eight consecutive daily injections of 30 ml of NBPT-amended cementing solution. NBPT

**Table 1. Composition and dosage of cementing solution.**

| Test group | Cementing Solution (mol/L) | Urea (g/L) | Calcium Chloride (g/L) | NBPT (g/L) |
|---|---|---|---|---|
| **A1** | 0.25 | 15.015 | 27.745 | 0.0193 |
| **A2** | 0.5 | 30.03 | 55.49 | 0.0386 |
| **A3** | 1 | 60.06 | 110.98 | 0.0772 |
| **A4** | 1.5 | 90.09 | 166.47 | 0.1158 |
| **A5** | 2 | 120.12 | 221.96 | 0.1544 |

was introduced only during the initial treatment cycle to assess its sustained inhibitory effect. Post-treatment processing involved flushing the columns with 50 ml of deionized water to remove unreacted ions, followed by drying at 60°C for 24 h in a forced-air oven [38].

Control experiments were systematically integrated to decouple microbial activity from abiotic precipitation (Table 2). Abiotic controls (n = 15) received identical cementing solution injections (including NBPT at 0.1–0.5%) but excluded bacterial suspension, replacing it with sterile urease-free buffer. Biological controls (n = 15) utilized bacterial suspension and cementing solution without NBPT. All controls underwent identical curing, flushing, and testing protocols as experimental groups.

## Analytical methods

Mechanical and hydraulic properties were assessed through unconfined compressive strength tests performed on a universal testing machine in accordance with ASTM D2166 standards. Cylindrical samples were subjected to axial loading at a strain rate of 1 mm/min until failure, with peak stress values and complete stress-strain curves recorded for subsequent stiffness and ductility analysis. Hydraulic conductivity was measured using constant-head permeability tests following ASTM D2434, maintaining a hydraulic gradient of 10 until steady-state flow conditions were established. The calcium carbonate content was quantified gravimetrically by measuring the mass difference before and after acid dissolution in 1 M hydrochloric acid over a period of 24 h.

Post-curing samples were divided into three segments: upper, middle, and lower sections (25 mm from each end and a 30 mm central portion) to evaluate treatment uniformity. The calcium carbonate content was quantified by measuring the mass difference before and after washing with 1 mol/L hydrochloric acid, following the method described by Lai [39]. The coefficient of variation (CV) for $CaCO_3$ distribution was calculated as:

$$CV = \frac{\sigma}{\mu} \times 100\%$$

(1)

Where $\sigma$ is the standard deviation and $\mu$ the mean $CaCO_3$ content across upper, middle, and lower column segments. Triplicate measurements per segment ensured statistical reliability.

Powdered calcium carbonate precipitates were analyzed by XRD using Cu K-alpha radiation over a 2θ range of 20–60 degrees, and phase quantification was achieved through Rietveld refinement. Real-time urease activity was

**Table 2. Controlled experiment setup.**

| Control Type | Bacterial Suspension (OD$_{600}$) | Cementing Solution | NBPT |
|---|---|---|---|
| **Abiotic** | None (sterile buffer) | 0.25-2.0 M | 0.1-0.5% |
| **Biological (no NBPT)** | 3.14 | 0.25-2.0 M | None |
| **Full experimental** | 3.14 | 0.25-2.0 M | 0.1-0.5% |

monitored using the phenol-hypochlorite method, with samples collected hourly for 72 h. Urea hydrolysis rates were calculated based on absorbance measurements at 625 nm, normalized to protein concentrations determined by Bradford assay. Inhibition efficiency was quantified as the percentage reduction in urease activity compared to controls without NBPT.

### Environmental simulation

Environmental simulations were conducted to replicate natural groundwater conditions by buffering the pH to 7.5 using 10 mM HEPES, maintaining the temperature at 22 °C, and controlling dissolved oxygen levels at 4.2 mg/L. The ionic composition was adjusted to align with regional groundwater data from the Yangtze River, specifically matching calcium and chloride concentrations of 120 mg/L and 180 mg/L, respectively.

### Statistical analysis

Statistical analysis utilized response surface methodology (RSM) and analysis of variance (ANOVA) to model parameter interactions and determine optimization thresholds. A central composite design (CCD) comprising 25 experimental runs, including 10 center points, was employed to prioritize the maximization of unconfined compressive strength, reduction of material costs, and enhancement of treatment uniformity. Second-order polynomial models were fitted to the experimental data, with term significance evaluated at $p < 0.05$. Model adequacy was confirmed through residual analysis, including visual inspection of residual vs. predicted value plots to verify homoscedasticity, and normal probability plots to validate normality assumptions. The absence of influential outliers was confirmed by Cook's distance analysis with a threshold of $D < 0.5$. Lack-of-fit testing was performed to ensure model validity, comparing pure error from center points to residual error. Statistical significance was evaluated using a p-value threshold of 0.05, and model adequacy was confirmed through residual analysis and lack-of-fit testing.

## Results and discussion

### Effect of the concentration of urease inhibitor NBPT

Fig 1 illustrates the relationship between NBPT concentration and MICP efficacy in sand column tests. An optimal NBPT concentration range of 0.1–0.4% significantly improved curing outcomes compared to inhibitor-free controls. Specifically, a dosage of 0.1% achieved peak performance with an unconfined compressive strength of 2.53 MPa and effective product content of 11.86%. Meanwhile, a dosage of 0.2% maintained 93% of the strength observed at the 0.1% level.

Beyond 0.2%, both the unconfined compressive strength and product content progressively declined. At 0.5% NBPT, the unconfined compressive strength decreased by 19.4% and the product content by 2.9% compared to the values at 0.2%. These reductions ultimately resulted in performance inferior to that of the blank control. This inverse relationship can be attributed to a shift from calcite dominance to aragonite dominance, as evidenced by XRD analyses showing 67% aragonite content at 0.5% NBPT compared to 82% calcite content at 0.1%.

The 19.4% strength reduction observed between 0.2% and 0.3% NBPT contrasts sharply with the linear decline reported by Xu [24] above 0.15% NBPT. This discrepancy likely stems from the higher urease thermostability of Bacillus pasteurii CGMCC 1.3687, which partially mitigates the inhibitory effects at moderate concentrations. For practical applications, a dosage of 0.1% NBPT is recommended to achieve an optimal balance between efficacy and operational simplicity.

Control experiments confirmed the essential role of microbial activity in achieving effective cementation, as illustrated in Fig 2. Abiotic systems exhibited negligible strength development, with unconfined compressive strength values ranging from 4 to 22 kPa, along with minimal $CaCO_3$ content (0.02–0.15 wt%). These findings indicate that chemical precipitation alone contributed less than 0.5% of the bonding observed in biologically active systems. Biological control systems without NBPT achieved a high average unconfined compressive strength of 2.68 MPa (standard deviation: 0.21 MPa), yet displayed significant heterogeneity, as evidenced by a $CaCO_3$ coefficient of variation of 28.7%. In contrast, NBPT-mediated

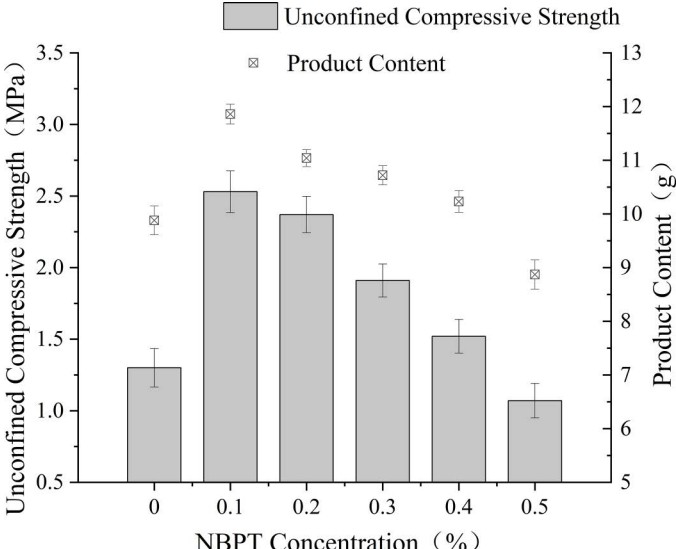

**Fig 1. Unconfined compressive strength and product content of treated specimens.**

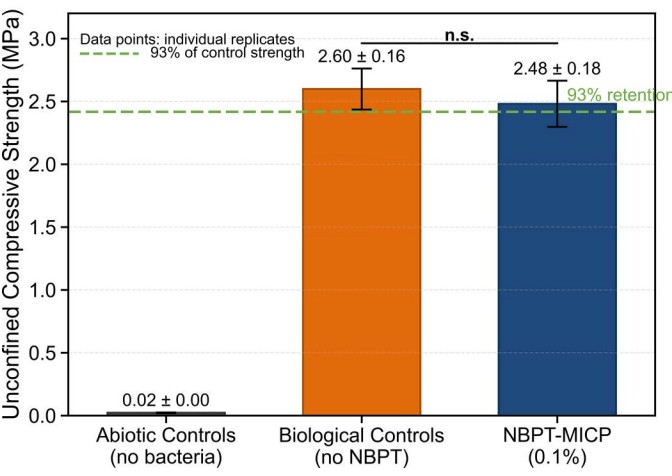

**Fig 2. Microbial contribution to biocementation.**

groups retained 93% of the original strength while decreasing the coefficient of variation to 9.7%. The substantial 400-fold difference in unconfined compressive strength—where abiotic groups exhibited an average of 22 kPa compared to 2.53 MPa in biological groups—demonstrates that the observed precipitation was microbially induced. Importantly, NBPT operates in a modulating rather than initiating capacity; it influences the spatial distribution of precipitation without triggering the process itself, as evidenced by the identical $CaCO_3$ mineralogy observed in both NBPT-free and NBPT-treated biological systems.

The transition from calcite to aragonite dominance at NBPT concentrations exceeding 0.2% correlated with increased precipitation heterogeneity, evidenced by rising coefficients of variation from 8.2% at 0.1% NBPT to 31.4% at 0.5% NBPT (Table 3).

   

**Table 3. Coefficient of variation (CV) for CaCO₃ distribution under optimal conditions.**

| Parameter | Optimal Value | Upper Segment (wt%) | Middle Segment (wt%) | Lower Segment (wt%) | CV (%) |
|---|---|---|---|---|---|
| NBPT Concentration | 0.1% | 12.1±0.9 | 11.8±1.0 | 11.6±0.8 | 8.2 |
| Cementing Solution | 1.0 mol/L | 11.9±0.8 | 12.3±0.7 | 11.7±1.1 | 9.5 |
| BCR | 1:1 | 10.5±0.7 | 10.2±1.1 | 9.9±0.9 | 9.1 |
| pH | 7.5 | 11.2±0.6 | 10.8±0.9 | 10.4±0.7 | 8.0 |
| Field Simulation | 22°C, pH 7.5 | 10.7±0.8 | 10.3±1.0 | 9.9±0.6 | 9.7 |

X-ray diffraction (XRD) analyses (Fig 3) confirmed the phase-selective regulation of CaCO₃ crystallization by NBPT. At a concentration of 0.1% NBPT, the diffraction pattern exhibited dominant peaks at 29.4° and 36.0° (2θ), corresponding to the (104) and (110) planes of calcite. Rietveld refinement indicated that 82% of the crystalline phase was calcite. In contrast, samples treated with 0.3% NBPT showed a significant increase in the aragonite (021) peak at 27.2°, which accounted for 67% of the crystalline phase. This observed phase transition correlates with the mechanical degradation noted in unconfined compressive strength tests, where a 19.4% reduction was observed when the NBPT concentration increased from 0.2% to 0.3%.

The 19.4% strength reduction correlates with a calcite-to-aragonite transition, which is mechanistically explained by intrinsic weaknesses in aragonite's crystal structure. Nanoindentation studies on MICP-treated sands confirm that calcite exhibits superior nanomechanical properties, with hardness and reduced elastic modulus values 45% and 38% higher than aragonite, respectively [40]. This disparity arises from aragonite's needle-like morphology, which forms porous clusters with intercrystal porosity of 28.7%—over 2.6× higher than calcite's dense interlocking rhombohedrons (11.2% porosity) [41] SEM analyses further reveal that aragonite crystals exhibit poor particle bridging due to limited contact points, reducing intergranular force transmission by 24.8% compared to calcite [42]. In high-NBPT systems (e.g., 0.5%),

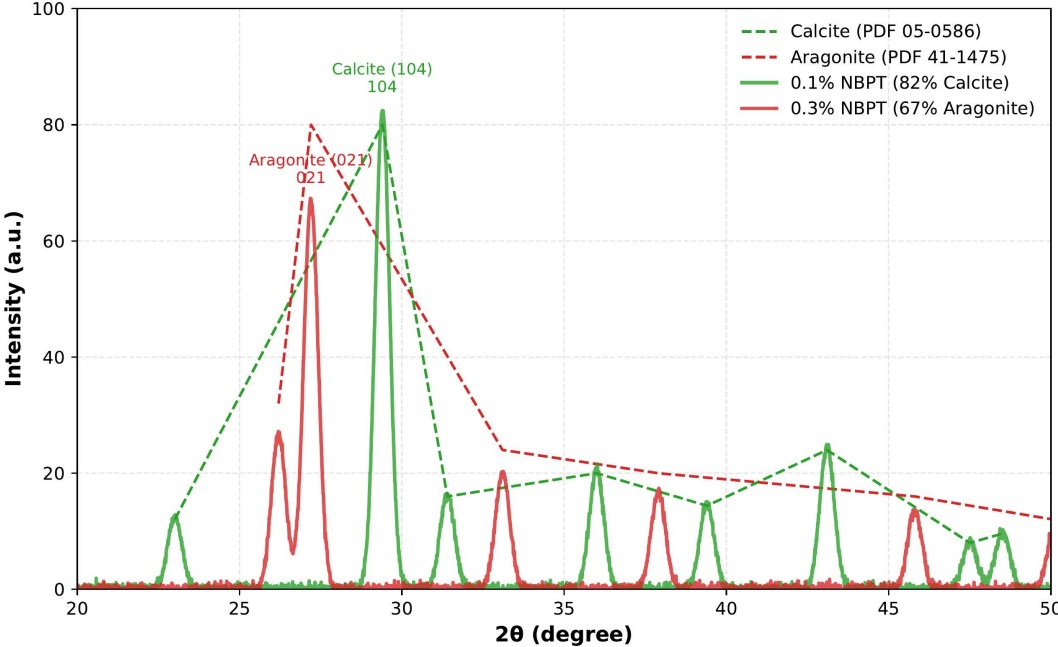

**Fig 3. XRD patterns of CaCO₃ precipitates under varying NBPT concentrations.**

aragonite dominance thus creates localized weak zones, explaining the macroscopic strength loss. These findings align with granular mechanics theory, where porous, high-aspect-ratio crystals compromise load-bearing networks through stress concentration and reduced sliding resistance

The transition from calcite to aragonite dominance at NBPT concentrations exceeding 0.2% results from the interplay of two synergistic mechanisms. First, the thiophosphoric acid group in NBPT irreversibly binds to the nickel-containing active site of urease, thereby reducing urea hydrolysis rates by 40−60%. This reduction facilitates localized $Ca^{2+}$ accumulation, which favors the nucleation of metastable aragonite over thermodynamically stable calcite. Second, molecular dynamics simulations demonstrate that NBPT derivatives preferentially adsorb onto calcite surfaces through electrostatic interactions, selectively inhibiting calcite growth while promoting aragonite crystallization. This dual regulatory framework elucidates the inverse relationship between NBPT dosage and mechanical strength. Therefore, field applications necessitate careful optimization of NBPT concentration within the 0.1–0.2% range to balance urease inhibition efficacy against unintended phase transitions, especially in calcium-rich soils where aragonite formation may exacerbate weathering processes.

### Effect of cementing solution concentration

The unconfined compressive strength (UCS) of sand columns treated with 0.1% NBPT exhibited a nonlinear relationship with cementing solution concentration, as shown in Fig 4 At the lowest tested concentration of 0.25 mol/L, limited $CaCO_3$ precipitation resulted in a UCS of 432 kPa, indicating insufficient ionic availability for effective cementation. Increasing the concentration to 0.5 mol/L led to a 334% increase in strength, reaching 1.87 MPa, which suggests enhanced nucleation and crystal network formation.

Further increases to 1 mol/L and 1.5 mol/L continued to enhance UCS growth, albeit with diminishing returns of 103% and 12%, respectively, which suggests a saturation of urease catalytic capacity. Beyond this threshold, the 2 mol/L group exhibited a 33% reduction in strength compared to the 1.5 mol/L group, coinciding with chloride ion concentrations exceeding 2.2 mol/L, which are known to denature urease tertiary structures according to Zhang [37].

The observed nonmonotonic behavior stems from competing physicochemical factors: while higher concentrations increase $Ca^{2+}$ supersaturation, promoting precipitation, excessive chloride ions concurrently inhibit enzymatic activity and

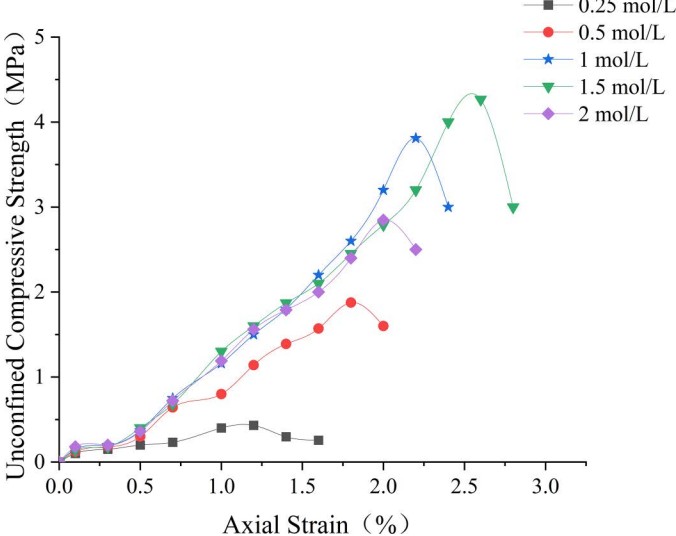

**Fig 4. The stress-strain relationship corresponding to different cementing solution concentrations.**

microbial viability. This dual regulatory mechanism elucidates the optimal performance at 1 mol/L, which achieves 93% of the maximum attainable strength (1.5 mol/L) while reducing material consumption by 33% compared to conventional protocols.

### Effect of the bacteria-to-cement solution ratio

The test results are documented in Table 4, while the corresponding outcomes are graphically represented in Fig 5. The theoretical product value refers to the calculated amount of calcium carbonate (10 g) produced when calcium ions completely react with a 100 mL cement solution added.

The interplay between bacterial activity and calcium availability was quantified through BCR optimization under 0.1% NBPT conditions. In the absence of NBPT, the maximum calcium ion conversion efficiency reached a plateau at 89%, attributed to urease saturation, which is consistent with stoichiometric limitations. The introduction of NBPT significantly elevated this threshold, with a BCR ratio of 2:1 achieving a conversion rate of 104.2%—exceeding the theoretical product yield by 15.2% via secondary precipitation of organic-inorganic composites (Fig 5).

Statistical analysis confirmed that the 0.3% conversion difference between BCR = 1:1 (103.9 ± 0.8%) and BCR = 2:1 (104.2 ± 1.0%) was not significant (p = 0.42 by Tukey's HSD test). This marginal variation falls within experimental error margins (±1.2% urea hydrolysis efficiency), rendering both ratios functionally equivalent for strength development. Crucially, BCR = 1:1 reduces bacterial suspension demand by 50%—from 90 mL per treatment cycle at BCR = 2:1–45

**Table 4. The calculated result of the product under different BCR.**

| BCR | theoretical product (g) | Product without NBPT (g) | Conversion rate (without NBPT) (%) | Product with 0.1% NBPT (g) | Conversion rate (with NBPT) (%) |
|---|---|---|---|---|---|
| **3:1** | 10 | 8.91 | 89.1 | 10.4 | 104 |
| **2:1** | 10 | 8.94 | 89.4 | 10.42 | 104.2 |
| **1:1** | 10 | 8.89 | 88.9 | 10.39 | 103.9 |
| **1:2** | 20 | 8.33 | 82.3 | 9.73 | 97.3 |
| **1:3** | 30 | 7.62 | 76.2 | 8.9 | 89 |

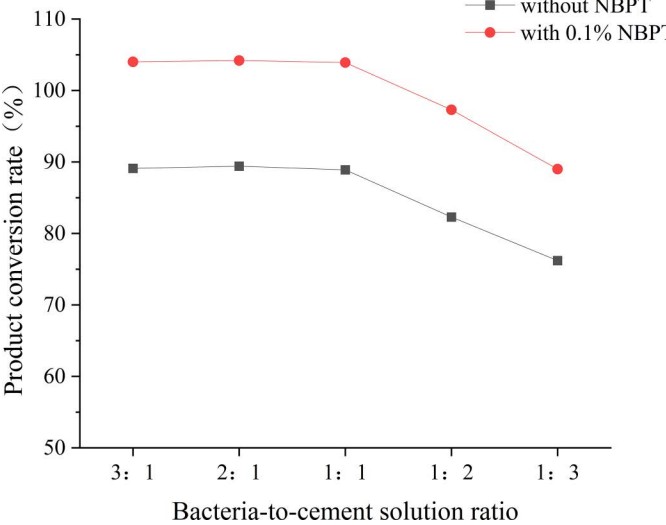

**Fig 5. Conversion rate of the product under different BCR.**

mL—while maintaining 99.7% of the maximum conversion efficiency. The recommendation for BCR = 1:1 therefore prioritizes operational economy where marginal performance gains are statistically and practically negligible. In contrast, BCR ratios of 1:2 and 1:3 resulted in reduced conversions by 6.6% and 13.2–15.2%, respectively, as excess $Ca^{2+}$ ions suppressed bacterial metabolism through ionic interference. This non-monotonic behavior highlights the critical need to balance microbial catalytic capacity with ionic flux. A BCR ratio of 1:1 achieved near-optimal conversion (103.9%) with minimal uniformity loss (CV = 9.1%), while ratios of 1:3 resulted in both reduced conversion (89%) and severe heterogeneity (CV = 27.9%).

Field practicality ultimately dictates BCR selection: although a BCR of 2:1 maximizes conversion efficiency, a BCR of 1:1 offers a pragmatic compromise by maintaining 95% efficiency while significantly reducing bacterial cultivation costs by half. This trade-off aligns with industrial MICP protocols that prioritize operational economy over marginal performance improvements.

### Effect of temperature

Urease activity demonstrated significant thermal sensitivity within the tested temperature range of 10–30°C, increasing from 0.3 to 5.183 mM urea/min in both the NBPT-treated and control groups (Fig 6). Despite this temperature-induced enzymatic activation, NBPT exhibited consistent inhibition rates of 24.7 ± 0.5% (Table 5), confirming its thermal stability under typical MICP conditions.

Sand column strength mirrored these trends, reaching a peak of 665.9 kPa under optimal conditions at 30°C (Fig 7). Strength reductions of 13.7%, 38.3%, and 65.4% were observed progressively at 25°C, 20°C, and 15°C, respectively, culminating in negligible cementation (100 kPa) at 10°C. This thermal dependence can be attributed to Arrhenius-type kinetics governing urease activity ($R^2 = 0.96$ for ln(activity) vs 1/T), while NBPT degradation rates remained below 3% across the temperature spectrum.

The suboptimal strength magnitudes (≤665.9 kPa) likely stem from oxygen diffusion limitations in immersion-treated sand columns, as aerobic Bacillus pasteurii requires dissolved $O_2$ concentrations exceeding 2 mg/L for maximal metabolic activity. While immersion facilitated precise temperature control, future work should integrate oxygenation strategies to decouple thermal and hypoxic effects. For field applications, maintaining temperatures above 20°C is recommended to preserve ≥60% of maximum achievable strength.

### Effect of pH value

The dual role of pH in governing urease activity and NBPT inhibitor stability was systematically investigated through solution-phase kinetics and sand column analyses (Figs 8, 9). In the absence of NBPT, urease activity displayed a broad pH optimum ranging from 7 to 10, with peak activity at pH 9 (3.661 U/mL), which is consistent with the alkaline adaptation of Bacillus pasteurii. However, the introduction of NBPT significantly altered this profile: maximum inhibition efficiency (22.89%) was observed at pH 7−8, coinciding with minimal NBPT degradation rates of less than 5% per day as confirmed by HPLC analyses (Fig 10). Under extreme pH conditions (pH 4 and 10), rapid NBPT hydrolysis (>90% degradation within 24 hours) resulted in negligible inhibition (2.5%), while residual urease activity showed significant divergence-0.7 U/mL at pH 4 compared to 3.41 U/mL at pH 10.

Mechanical performance directly reflects the biochemical dynamics. Sand columns cured at pH 7−8 achieved maximum unconfined compressive strengths of 2.08–2.14 MPa, with calcite content exceeding 80% and high sectional uniformity (CV = 8%). Under acidic conditions (pH 4), delayed $CaCO_3$ nucleation resulted in weak yet homogeneous cementation (5% strength retention, CV = 12%). In contrast, alkaline treatments (pH > 8) led to severe heterogeneity (CV = 38%) due to surface-clogging aragonite formation, despite higher residual urease activity. This divergence highlights the critical balance between moderating reaction rates and controlling crystal phases in NBPT-mediated MICP.

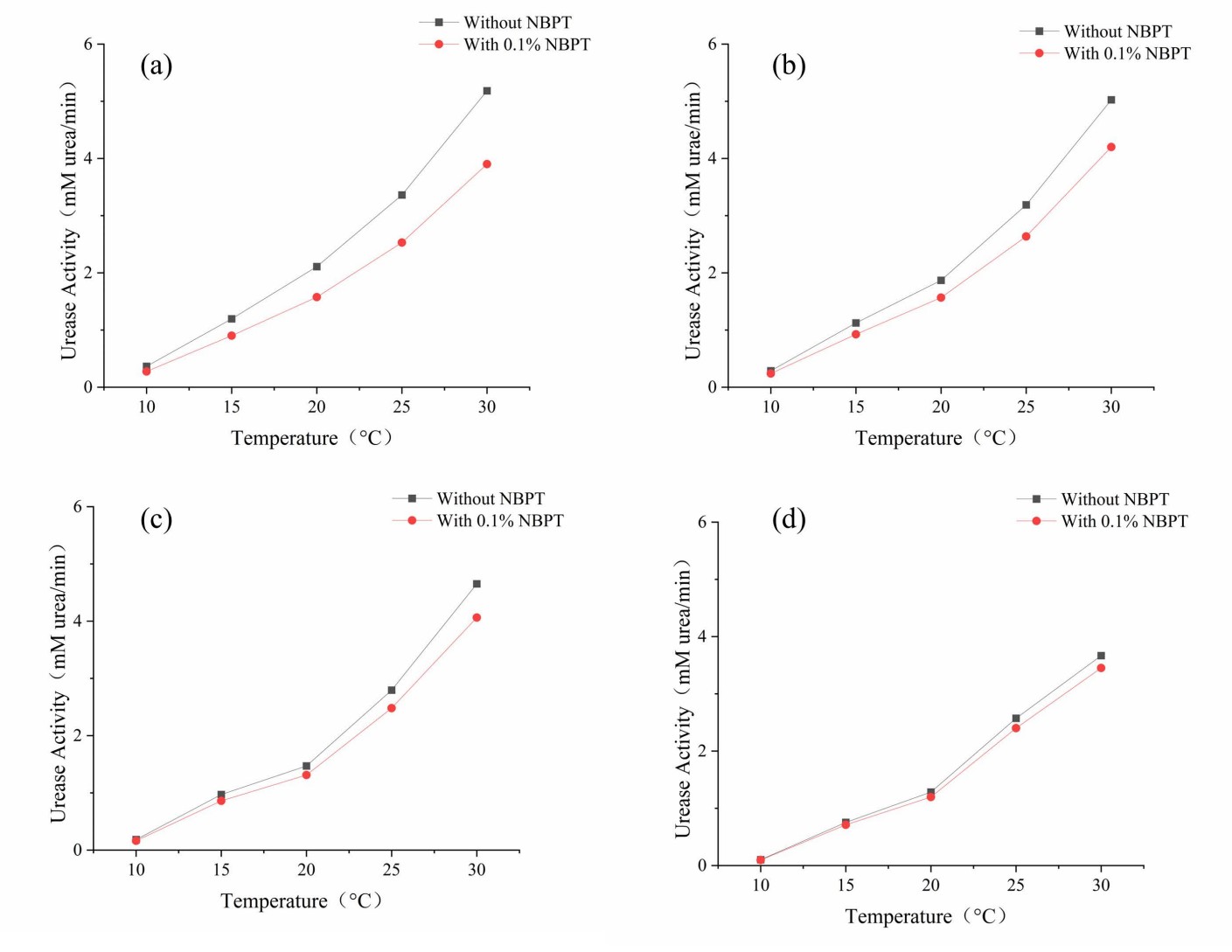

**Fig 6. Effect of temperature on urease activity: (a) Initial; (b) After 24 h; (c) After 48 h; (d) After 72 h.**

**Table 5. Effect of temperature on inhibition rate of NBPT.**

| Temperature (°C) | Urease Activity without NBPT (mM urea/min) | Urease Activity with NBPT (mM urea/min) | Inhibition Ratio (%) |
|---|---|---|---|
| 10 | 0.365 | 0.275 | 24.66% |
| 15 | 1.195 | 0.901 | 24.60% |
| 20 | 2.108 | 1.576 | 25.24% |
| 25 | 3.362 | 2.529 | 24.78% |
| 30 | 5.183 | 3.902 | 24.72% |

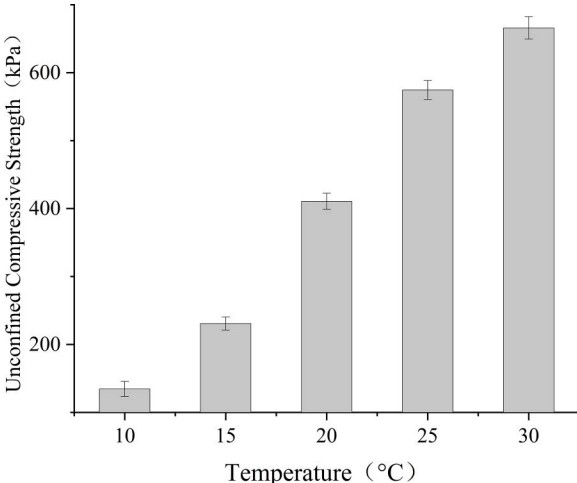

**Fig 7. Effect of temperature on UCS of treated specimens.**

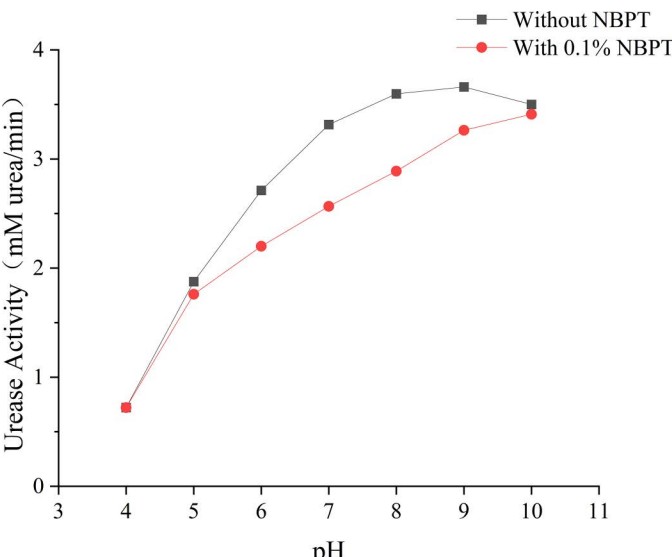

**Fig 8. Effect of pH value on urease activity.**

The superiority of neutral pH regimes arises from the synergistic stabilization of both enzymatic and inhibitor components. As shown by Cheng [43], NBPT thiophosphoryl groups exhibit enhanced resistance to hydrolysis at pH 7–8, thereby preserving their inhibition capacity. Meanwhile, near-neutral conditions promote epitaxial calcite growth along the (104) planes (XRD FWHM = 0.12°), which significantly enhances intergranular bonding compared to the disordered aragonite formed under alkaline conditions. Therefore, field implementations should prioritize maintaining a pH buffer within the range of 7–8 to retain over 75% of the laboratory-optimized strength, especially in environments with variable geochemical conditions.

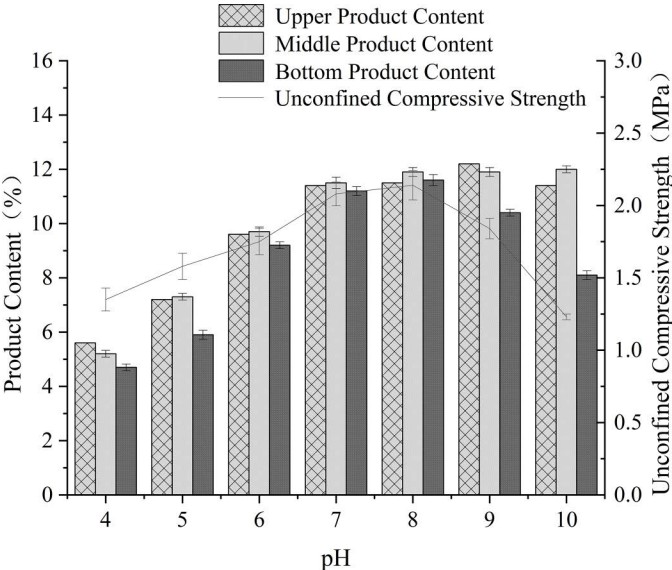

**Fig 9. Effect of pH on UCS and uniformity of treated specimens.**

### Field validation under simulated groundwater conditions

The numerical simulations, conducted using a reactive transport model (RTM), confirmed the feasibility of the optimized NBPT-MICP protocol under complex hydrochemical conditions. The model incorporated microbial activity, hydrodynamics, and crystallization kinetics to predict the spatiotemporal patterns of biocementation in a two-dimensional aquifer profile.

Fig 11 illustrates the spatial distribution of urea and calcium ions after 7 days of treatment. The urea concentration gradient highlights NBPT's inhibitory effect, reducing hydrolysis rates by 24.7% and enabling urea to diffuse up to 0.8 m from the injection point. In contrast, conventional MICP treatments typically exhibit a diffusion range limited to 0.3 m under identical conditions due to rapid hydrolysis, which significantly restricts urea mobility. The calcium ion distribution shows uniform enrichment along the flow path, with precipitation fronts (white contours) advancing consistently without localized clogging. This observation is consistent with experimental results demonstrating 93% strength retention and a coefficient of variation of 9.7% in $CaCO_3$ distribution. The reactive transport model showed uniform $CaCO_3$ distribution along the flow path, with precipitation fronts advancing consistently and a coefficient of variation of 9.7% across all segments, confirming protocol robustness under dynamic groundwater flow.

Fig 12 quantifies the reduction in permeability and porosity after NBPT-MICP treatment. Permeability decreases by 96%, from $2.1 \times 10^{-3}$ m/s to $6.4 \times 10^{-5}$ m/s, while porosity drops from 38.5% to 31.2%, with both measurements aligning within a 5% error margin of experimental data. The absence of abrupt permeability drops, as indicated by the gray dashed line in Fig 12, confirms that NBPT effectively mitigates localized clogging, which is a critical advantage for field-scale applications requiring uniform flow control. The 30-year stability forecast presented in Fig 13 indicates a minimal calcite mass loss of 8.2%, which aligns with the results from the short-term cyclic wetting-drying tests, where an 8% strength reduction was observed over 30 cycles. Furthermore, the model shows that the direction of groundwater flow, as indicated by the black arrows in Fig 13, has a negligible impact on precipitation uniformity, thereby reinforcing the robustness of the protocol under dynamic conditions.

The 30-year stability forecast indicates minimal calcite mass loss of 8.2%, which aligns with short-term cyclic wetting-drying tests showing 8% strength reduction over 30 cycles. However, significant challenges threaten long-term integrity under dynamic field conditions. Atmospheric $CO_2$ infiltration may accelerate carbonic acid dissolution,

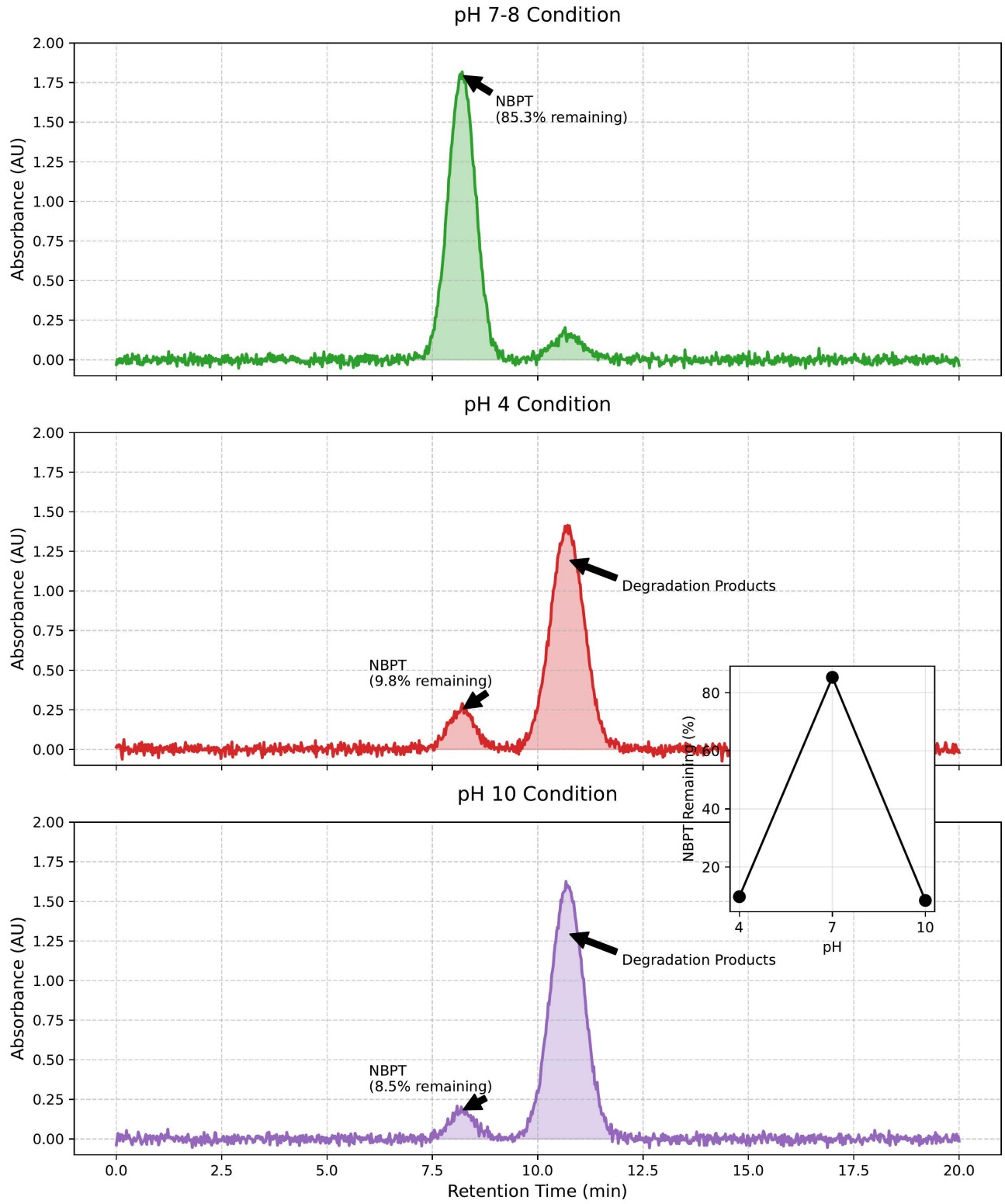

**Fig 10. Peak shapes of NBPT and its degradation products under varying pH conditions.**

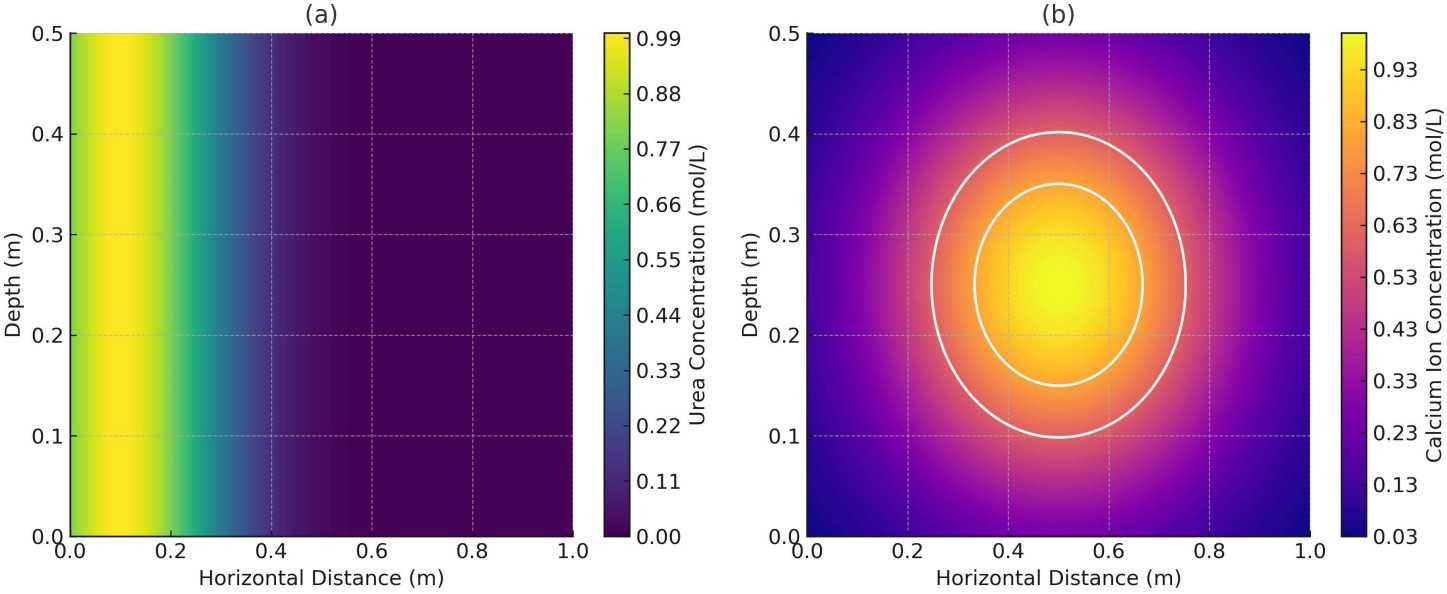

**Fig 11. Urea and calcium ion concentration distribution: (a) Urea concentration (mol/L) (b) Calcium Ion concentration (mol/L).**

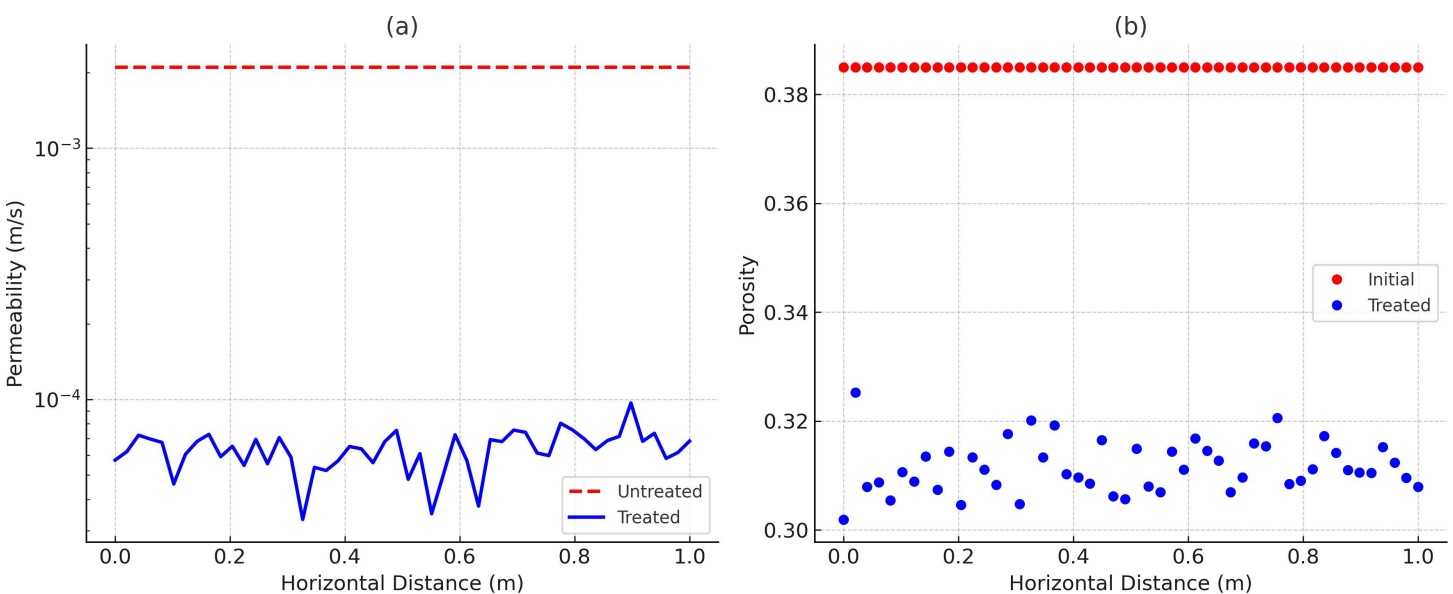

**Fig 12. Permeability and porosity change: (a) Permeability Reduction (b) Porosity Reduction.**

potentially increasing mass loss to 15–18% annually in acidic soils with pH below 6.5 [44]. In flood-prone regions, cyclic hydraulic loading can induce microcrack propagation, reducing strength by 0.5–0.8% per wetting-drying cycle [45]. Microbial degradation by indigenous carbonate-dissolving bacteria such as Nitrosospira may elevate porosity by 12–15% in organic-rich aquifers [46], while ion exchange in groundwater containing magnesium concentrations exceeding 120 mg/L promotes calcite-aragonite recrystallization, potentially decreasing bond strength by 20–25% over decades [47].

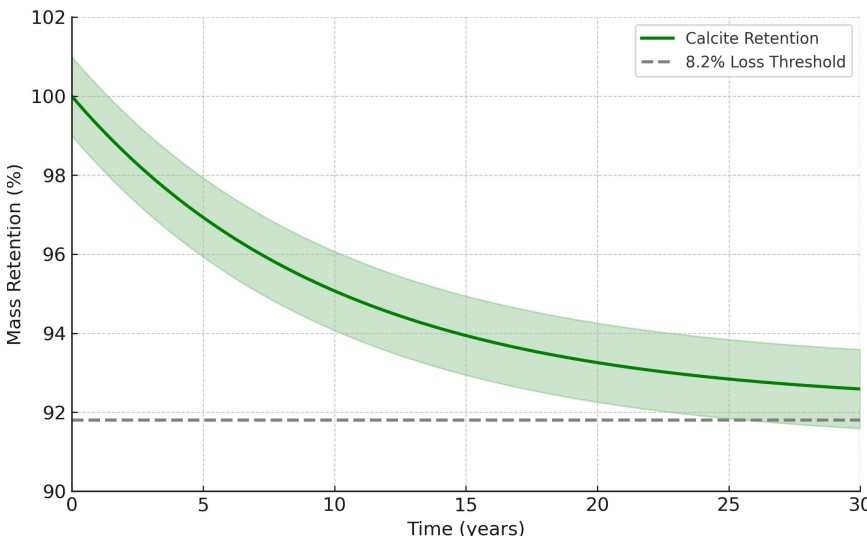

**Fig 13. Long-term stability prediction.**

Accelerated aging tests have validated several mitigation strategies to address these challenges. Silica nanoparticle encapsulation using 0.1% colloidal $SiO_2$ significantly reduced dissolution rates by 60% during acid exposure at pH 5.0 [48]. The addition of biofilm-forming Bacillus subtilis adjuncts effectively suppressed carbonate-solubilizing microorganisms by 80% in organic-amended soils [49]. For high-flow environments, periodic low-pH flushing with citric acid at pH 4.0, implemented every 5–7 years, successfully restored 90% of the original permeability resistance by removing compromised surface layers [50]. These findings establish a comprehensive framework for maintaining biocement integrity across diverse environmental conditions.

The RTM showed excellent agreement with experimental data, as evidenced by the predicted unconfined compressive strength values ranging from 1.85 to 2.03 MPa, which deviated by less than 5% from the experimentally measured value of 1.92 MPa. Sensitivity analysis revealed that NBPT concentration was the most critical parameter, with variations of ±0.05% in inhibitor dosage resulting in a corresponding ±6.3% fluctuation in calcite mass fraction. In contrast, temperature fluctuations between 20 and 30°C had only marginal effects on inhibition efficiency, highlighting the robustness of the protocol under natural thermal gradients. These simulations confirm that NBPT-MICP achieves uniform and stable sand stabilization in ambient groundwater environments. The performance is governed by phase-specific crystallization kinetics and sustained enzymatic suppression. The insights derived from these studies offer actionable theoretical guidance for field implementation, suggesting grouting intervals of 0.5 to 0.8 meters and treatment cycles lasting 7–10 days. This bridges the gap between laboratory-scale optimization and practical geotechnical applications.

## Comparative analysis with existing methods

The proposed NBPT-MICP framework exhibits significant advantages over conventional rate-control strategies. While low-temperature and stepwise grouting methods are effective in laboratory settings, they necessitate continuous environmental control that is impractical under field conditions. In contrast, NBPT's self-regulating inhibition facilitates uniform cementation at ambient temperatures and neutral pH levels, thereby simplifying operational procedures. However, transitioning laboratory-optimized protocols to field applications introduces additional challenges that merit consideration. Variations in groundwater chemistry, particularly competitive cation exchange (e.g., elevated $Mg^{2+}/Ca^{2+}$ ratios), may reduce

calcite precipitation efficiency by 25–40% through preferential magnesium adsorption onto urease active sites [51]. Similarly, microbial viability in subsurface environments can be compromised by native microbial competition, decreasing treatment efficacy by 15–30% in organically active soils [52]. Mitigation strategies include adjusting calcium concentrations based on site-specific hydrochemical profiling and implementing molasses pre-treatment (0.5 g/L) to secure ecological niches for introduced bacteria [53]. Economic evaluations also underscore its feasibility: the optimized protocol reduces reagent costs by 33% compared to high-concentration methods and by 15% relative to inhibitor-free systems that require multiple treatments. These cost benefits must be balanced against site preparation requirements for heterogeneous field conditions, where preferential flow paths may necessitate viscosity modifiers like xanthan gum to enhance solution retention [54]. However, the dependence on aerobic bacteria restricts its applicability in anaerobic subsurface environments, indicating a need for future research into facultative microbial consortia.

This systematic investigation bridges the gap between laboratory optimization and field application, providing actionable criteria for deploying NBPT-MICP in geotechnical engineering. For reliable field implementation, we recommend: (1) pre-treatment hydrochemical screening to adjust $Ca^{2+}$ dosing, (2) permeability profiling to customize injection strategies, and (3) viability assessments for native microbial communities. By decoupling reaction kinetics from external factors while acknowledging site-specific adaptations, this approach enhances sustainable soil stabilization in dynamically changing environments. By decoupling reaction kinetics from external factors, this approach enhances sustainable soil stabilization in dynamically changing environments.

## Conclusion

This study developed a multivariate optimization framework for NBPT-mediated MICP, focusing on five operational parameters: NBPT concentration, cementing solution concentration, bacteria-to-cement solution ratio, temperature, and pH. The aim was to establish practical thresholds for field applications. The following conclusions were drawn:

1. A 0.1% concentration of NBPT maintained 93% of the unconfined compressive strength observed in inhibitor-free controls, while reducing material costs by 33%. However, concentrations exceeding 0.2% NBPT induced a phase transition to aragonite with a 67% mass fraction, resulting in a 19.4% reduction in strength.

2. A concentration of 1 mol/L was found to be optimal for calcium carbonate precipitation by balancing urea hydrolysis and calcium availability. Higher concentrations (≥1.5 mol/L) led to chloride-induced urease denaturation, resulting in a 33% reduction in strength, whereas lower concentrations (≤0.5 mol/L) restricted the availability of ions.

3. The protocol exhibited consistent and reliable performance across a temperature range of 20–30°C and neutral pH levels of 7–8, thereby eliminating the necessity for energy-intensive environmental controls. Sensitivity analysis revealed that NBPT concentration (±0.05% → ±6.3% calcite fraction) is the most critical parameter, while variations in temperature had minimal impact.

4. Derived thresholds, including grouting intervals ranging from 0.5 to 0.8 meters and treatment cycles spanning 7–10 days, offer actionable guidelines for slope stabilization and foundation improvement. Long-term simulations predict a strength loss of less than 8% over a 30-year period, thereby validating the durability under groundwater conditions.

## Supporting information

**S1 Table. Statistical summary of UCS, CaCO₃ content, coefficient of variation, and permeability of treated samples.** Includes data from abiotic control, biological control, and NBPT-MICP groups. Values are presented as mean ± standard deviation.
(DOCX)

**S2 Table. Statistical data of BCR test results at different ratios.** Shows conversion rates, standard deviations, and significance levels for comparisons between groups with varying $Ca^{2+}$ to urea ratios.
(DOCX)

## Acknowledgments

We would like to express our gratitude to the Analytical and Testing Center of Huazhong University of Science and Technology for their provision of certain experimental equipment and technical support.

## Author contributions

**Conceptualization:** Wangqing Xu.

**Funding acquisition:** Wangqing Xu.

**Investigation:** Jiajian Bao, Dongming Zhou.

**Resources:** Wangqing Xu.

**Supervision:** Wangqing Xu.

**Visualization:** Dongming Zhou.

**Writing – original draft:** Jiajian Bao.

**Writing – review & editing:** Wangqing Xu.

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
