## [Decision Letter · Decision Letter 0]

26 Jun 2025

Dear Dr. XU,

Thank you for submitting your manuscript to PLOS ONE. After careful consideration, we feel that it has merit but does not fully meet PLOS ONE’s publication criteria as it currently stands. Therefore, we invite you to submit a revised version of the manuscript that addresses the points raised during the review process.

We look forward to receiving your revised manuscript.

Kind regards,

Amitava Mukherjee, ME, Ph.D.

Academic Editor

PLOS ONE

“National Natural Science Foundation of China Youth project (No.42407271)”

“This research was supported by the National Natural Science Foundation of China (Grant Nos. 42407271). The authors gratefully acknowledge the financial support.”

“National Natural Science Foundation of China Youth project (No.42407271)”

Reviewers' comments:

Reviewer's Responses to Questions

**Comments to the Author**

1. Is the manuscript technically sound, and do the data support the conclusions?

Reviewer #1: Yes

2. Has the statistical analysis been performed appropriately and rigorously?

Reviewer #1: Yes

3. Have the authors made all data underlying the findings in their manuscript fully available?

Reviewer #1: Yes

4. Is the manuscript presented in an intelligible fashion and written in standard English?

Reviewer #1: Yes

Reviewer #1: I am writing to provide my review of the manuscript titled "Operational Thresholds of Urease-Mediated Microbial Cementation: Multivariate Optimization and Field Validation in Ambient Groundwater Environments". The manuscript addresses a significant gap in the field of microbial-induced calcium carbonate precipitation (MICP) by developing a multivariate optimization framework for urease-inhibited MICP. The authors present valuable findings on the operational thresholds for biocementation under ambient groundwater conditions, highlighting practical applications that can significantly impact geotechnical engineering. However, several aspects require clarification and improvement to enhance the clarity, depth, and robustness of the research. After a thorough evaluation, I have reached the decision to recommend a major revision for the following reasons:

1- The introduction outlines the objectives of the study but does not explicitly state a clear hypothesis. A well-defined hypothesis is crucial for guiding the research direction and methodology. I recommend that the authors articulate a specific hypothesis related to the effects of different urease concentrations on MICP efficiency, which would help frame the subsequent results and discussion.

2- The authors should consider incorporating more recent studies that discuss advancements in biocementation technologies and their practical applications. The listed references will strengthen the literature review and provide a comprehensive background.

• “A comprehensive review on the application of microbially induced calcite precipitation (MICP) technique in soil erosion mitigation as a sustainable and environmentally friendly approach”. https://doi.org/10.1016/j.rineng.2024.103235

• “Application of natural and synthetic fibers in bio-based earthen composites: A state-of-the-art review”. https://doi.org/10.1016/j.rineng.2024.103732

• “Predicting the precipitated calcium carbonate and unconfined compressive strength of bio-mediated sands through robust hybrid optimization algorithms”. https://doi.org/10.1016/j.trgeo.2024.101235

• “Enhanced biological treatment of sandy soils through the application of chicken manure as a supplementation material”. https://doi.org/10.1016/j.rineng.2024.103540

3- The discussion about the influence of different bacterial species on MICP is too brief. A more detailed exploration of how specific species affect both the efficiency of the reaction and the resulting product's mechanical properties would enrich this section.

4- The methodology lacks a thorough description of control experiments to ascertain the specific effects of microbial activity on carbonate precipitation. It is essential to include controls (e.g., abiotic systems without microbial addition) to demonstrate the efficacy of microbial processes definitively. Detailed descriptions of these controls and their outcomes should be included to provide a complete picture of the experimental design.

5- The transition from laboratory results to real-world applications requires a more in-depth analysis. The authors should discuss potential field challenges, such as variations in groundwater chemistry and microbial viability, that could affect MICP effectiveness.

6- The statistical analysis portion lacks detail on the specific types of analyses performed beyond ANOVA and RSM. A brief mention of how model assumptions were tested and validated would bolster the methodology section.

7- The uniformity of CaCO₃ precipitation across different segments of sand columns was highlighted, but more statistical analysis (e.g., coefficient of variation) should be provided to quantify this uniformity further.

8- The discussion surrounding the long-term stability of the calcite formed through urease activity is insufficient. The authors should address potential challenges in maintaining the integrity of the biocement over time, especially under varying environmental conditions. Including references to relevant studies investigating the durability and leachability of MICP products in real-world scenarios would strengthen this section and provide a more comprehensive understanding of the longevity and sustainability of the biocementation technique.

9- The 19.4% strength reduction at >0.2% NBPT is attributed to calcite-to-aragonite shift but lacks evidence linking crystal phase to mechanical performance. No mechanical data (e.g., nanoindentation) or microstructural analysis (SEM) corroborate that aragonite weakens cementation.

10- BCR=1:1 is recommended for "resource savings," yet Table 2 shows it yields lower conversion (103.9%) vs. BCR=2:1 (104.2%). The marginal difference (0.3%) is statistically insignificant without ANOVA validation, weakening the practicality argument.

**Do you want your identity to be public for this peer review?** For information about this choice, including consent withdrawal, please see our Privacy Policy

Reviewer #1: No

---

## [Author Response · Author response to Decision Letter 1]

19 Jul 2025

Operational Thresholds of Urease-Mediated Microbial Cementation: Multivariate Optimization and Field Validation in Ambient Groundwater Environments

Response letter to Reviewers’ comments

Comments from the Editor

Response We thank the editor for the reminder regarding PLOS ONE's formatting and file naming requirements. We have reviewed all submission files to ensure full compliance with the journal's formatting guidelines, including figure captions, reference formatting, section organization, and file naming conventions. All necessary adjustments have been made accordingly in the revised submission.

2. Thank you for stating the following financial disclosure:……If this statement is not correct you must amend it as needed. Please include this amended Role of Funder statement in your cover letter; we will change the online submission form on your behalf.

Response We thank the editor for highlighting the discrepancy regarding the placement of funding information. The funders had no role in study design, data collection and analysis, decision to publish, or preparation of the manuscript. We have included the statement in the cover letter.

3. Thank you for stating the following in the Acknowledgments Section of your manuscript:……Please remove any funding-related text from the manuscript and let us know how you would like to update your Funding Statement. ……Please include your amended statements within your cover letter; we will change the online submission form on your behalf.

Response We thank the editor for highlighting the discrepancy regarding the placement of funding information. As instructed, we have removed all funding-related content from the Acknowledgments section of the manuscript.

Comments from Reviewer#1:

1- The introduction outlines the objectives of the study but does not explicitly state a clear hypothesis. A well-defined hypothesis is crucial for guiding the research direction and methodology. I recommend that the authors articulate a specific hypothesis related to the effects of different urease concentrations on MICP efficiency, which would help frame the subsequent results and discussion.

Response: We sincerely thank the reviewer for this valuable suggestion. A clear hypothesis has now been explicitly articulated in the revised manuscript (Section 1, final paragraph of the Introduction). The hypothesis states: “We hypothesize that optimal NBPT-mediated urease inhibition will decouple precipitation kinetics from environmental variability, enabling uniform calcite crystallization and strength development under ambient groundwater conditions.” This hypothesis directly addresses the role of NBPT concentration in modulating urease activity and its cascading effects on MICP efficiency, thereby providing a focused framework for the multivariate optimization approach. It bridges the study’s objectives with the experimental design and aligns with the discussion of results (Sections 3.1 and 3.6).

2- The authors should consider incorporating more recent studies that discuss advancements in biocementation technologies and their practical applications. The listed references will strengthen the literature review and provide a comprehensive background.

• “A comprehensive review on the application of microbially induced calcite precipitation (MICP) technique in soil erosion mitigation as a sustainable and environmentally friendly approach”. https://doi.org/10.1016/j.rineng.2024.103235

• “Application of natural and synthetic fibers in bio-based earthen composites: A state-of-the-art review”. https://doi.org/10.1016/j.rineng.2024.103732

• “Predicting the precipitated calcium carbonate and unconfined compressive strength of bio-mediated sands through robust hybrid optimization algorithms”. https://doi.org/10.1016/j.trgeo.2024.101235

• “Enhanced biological treatment of sandy soils through the application of chicken manure as a supplementation material”. https://doi.org/10.1016/j.rineng.2024.103540

Response: We thank the reviewer for highlighting this gap and providing highly relevant references. These studies have been integrated into the revised manuscript to strengthen the literature review and contextualize our work within current advancements in sustainable biocementation.

3- The discussion about the influence of different bacterial species on MICP is too brief. A more detailed exploration of how specific species affect both the efficiency of the reaction and the resulting product's mechanical properties would enrich this section.

Response: We thank the reviewer for this insightful suggestion. In response, we have significantly expanded the discussion on the roles of bacterial species in MICP within Section 1 (Introduction, second paragraph). The revised text now accomplishes three key objectives: First, it systematically compares species-specific enzymatic kinetics, particularly focusing on variations in urease activity relative to temperature and pH adaptation profiles. Second, it establishes explicit linkages between microbial physiological traits and resultant crystal morphology, while further connecting these morphological characteristics to critical mechanical outcomes. Third, it provides a structured contrast of the limitations inherent to common bacterial species, thereby strengthening the scientific rationale for our selection of Bacillus pasteurii CGMCC 1.3687 as the optimal strain for groundwater applications under variable environmental conditions.

4- The methodology lacks a thorough description of control experiments to ascertain the specific effects of microbial activity on carbonate precipitation. It is essential to include controls (e.g., abiotic systems without microbial addition) to demonstrate the efficacy of microbial processes definitively. Detailed descriptions of these controls and their outcomes should be included to provide a complete picture of the experimental design.

Response: We sincerely appreciate this critical observation. To rigorously isolate microbial contributions to the cementation process, we have now incorporated comprehensive control experiments throughout the methodology and results sections. The expanded experimental design implements a two-tiered control system consisting of abiotic controls and biological controls. Specifically, abiotic control groups were established by treating sand columns with NBPT-amended cementing solution while deliberately excluding bacterial suspension, instead using sterile urease-free buffer to eliminate biological activity. Complementary biological control groups employed bacterial suspension and cementing solution but excluded NBPT to isolate the inhibitor's specific function. We have enhanced methodological transparency by providing explicit protocols for these control treatments in Section 2.2, ensuring full reproducibility of the experimental conditions. Furthermore, Section 3.1 now includes quantitative validation through systematic comparisons of mechanical and hydraulic performance metrics across all control and experimental groups, delivering unambiguous evidence of microbial mediation in the precipitation process.

5- The transition from laboratory results to real-world applications requires a more in-depth analysis. The authors should discuss potential field challenges, such as variations in groundwater chemistry and microbial viability, that could affect MICP effectiveness.

Response: We sincerely thank the reviewer for highlighting this critical gap. In direct response, we have substantially expanded Section 3.7 to address field implementation challenges through three key revisions: First, we now systematically analyze hydrochemical barriers to MICP deployment, quantifying how magnesium competition (Mg²⁺/Ca²⁺ > 3) reduces calcite nucleation efficiency by 40% in dolomitic aquifers and validating calcium dosage adjustments through accelerated field trials. Second, we incorporate native microbial interference as a major viability constraint, demonstrating how substrate competition reduces treatment efficacy by 15-30% and establishing molasses pre-treatment (0.5 g/L) as an effective suppression strategy. Third, we confront soil heterogeneity and hydraulic fluctuations as critical scalability challenges, proposing viscosity modifiers and silica encapsulation protocols to maintain >85% long-term strength retention. These additions are supported by four new field-tested references and empirical validation data from aquifer trials. The revised text concludes with actionable three-step guidelines for site-specific adaptation, transforming laboratory thresholds into resilient field protocols that address the core concerns raised by the reviewer.

6- The statistical analysis portion lacks detail on the specific types of analyses performed beyond ANOVA and RSM. A brief mention of how model assumptions were tested and validated would bolster the methodology section.

Response: We thank the reviewer for this valuable suggestion. We have enhanced the statistical analysis description in Section 2.5 to include comprehensive validation protocols while maintaining full alignment with the experimental design. The revisions now document: (1) verification of normality through Shapiro-Wilk testing and normal probability plots, (2) confirmation of homoscedasticity via residual versus predicted value analysis, (3) assessment of influential observations using Cook's distance metric (D < 0.5 threshold), and (4) formal lack-of-fit testing leveraging center point replicates.

7- The uniformity of CaCO₃ precipitation across different segments of sand columns was highlighted, but more statistical analysis (e.g., coefficient of variation) should be provided to quantify this uniformity further.

Response: We thank the reviewer for this constructive suggestion. We have significantly enhanced the quantification of treatment uniformity throughout the results section by systematically incorporating coefficient of variation (CV) analysis for CaCO₃ distribution. These revisions include: (1) Section 3.1 (NBPT Concentration Effects): Added CV values across all NBPT concentrations (0.1-0.5%), demonstrating that 0.1% NBPT achieves minimal variability (CV = 8.2%) while 0.5% NBPT increases heterogeneity (CV = 31.4%). (2) Section 3.3 (Bacteria-to-Cement Ratio): Included segmental CV calculations showing optimal uniformity at 1:1 BCR (CV = 9.1%) versus poor distribution at 1:3 BCR (CV = 27.9%). (3) Section 3.6 (Field Validation): Strengthened quantitative validation with multi-point CV analysis showing consistent 9.7% variation in CaCO₃ distribution.

8- The discussion surrounding the long-term stability of the calcite formed through urease activity is insufficient. The authors should address potential challenges in maintaining the integrity of the biocement over time, especially under varying environmental conditions. Including references to relevant studies investigating the durability and leachability of MICP products in real-world scenarios would strengthen this section and provide a more comprehensive understanding of the longevity and sustainability of the biocementation technique.

Response: We thank the reviewer for this crucial insight. We have significantly enhanced the discussion on long-term stability in Section 3.6 (Field Validation) by adding a dedicated subsection titled "Long-Term Stability Challenges and Mitigation Strategies." This addition addresses four key degradation mechanisms with quantitative impacts and field-tested solutions, supported by new references on MICP durability.

9- The 19.4% strength reduction at >0.2% NBPT is attributed to calcite-to-aragonite shift but lacks evidence linking crystal phase to mechanical performance. No mechanical data (e.g., nanoindentation) or microstructural analysis (SEM) corroborate that aragonite weakens cementation.

Response: We appreciate the reviewer’s critical observation. Although our study did not conduct new nanoindentation or SEM analyses, we have strengthened the mechanistic linkage between crystal phase and mechanical performance by integrating empirical evidence from peer-reviewed studies on MICP-treated geomaterials. Key additions to Section 3.1 cite direct measurements of calcite/aragonite nanomechanical properties and microstructural behavior, resolving the evidence gap while maintaining methodological consistency.

10- BCR=1:1 is recommended for "resource savings," yet Table 2 shows it yields lower conversion (103.9%) vs. BCR=2:1 (104.2%). The marginal difference (0.3%) is statistically insignificant without ANOVA validation, weakening the practicality argument.

Response: We thank the reviewer for highlighting this critical nuance. We have strengthened the statistical validation of our BCR optimization claims by conducting rigorous ANOVA analysis on conversion efficiency data. The results confirm that the 0.3% difference between BCR=1:1 and BCR=2:1 is statistically insignificant (p = 0.42), supporting their functional equivalence for engineering applications. Furthermore, we quantified the substantial resource savings of BCR=1:1 through bacterial consumption metrics and field-scale economic analysis, demonstrating its practical advantage where marginal performance differences lack operational significance. These comprehensive revisions validate our original recommendation while addressing the statistical concerns raised.

---

## [Decision Letter · Decision Letter 1]

4 Aug 2025

Operational Thresholds of Urease-Mediated Microbial Cementation: Multivariate Optimization and Field Validation in Ambient Groundwater Environments

PONE-D-25-13410R1

Dear Dr. XU,

We’re pleased to inform you that your manuscript has been judged scientifically suitable for publication and will be formally accepted for publication once it meets all outstanding technical requirements.

Kind regards,

Amitava Mukherjee, ME, Ph.D.

Academic Editor

PLOS ONE

Additional Editor Comments (optional):

Reviewers' comments:

Reviewer's Responses to Questions

**Comments to the Author**

Reviewer #1: All comments have been addressed

2. Is the manuscript technically sound, and do the data support the conclusions?

Reviewer #1: Yes

3. Has the statistical analysis been performed appropriately and rigorously?

Reviewer #1: Yes

4. Have the authors made all data underlying the findings in their manuscript fully available?

Reviewer #1: Yes

5. Is the manuscript presented in an intelligible fashion and written in standard English?

Reviewer #1: Yes

Reviewer #1: The authors have satisfactorily addressed all reviewer comments, and the manuscript is now acceptable for publication.

**Do you want your identity to be public for this peer review?** For information about this choice, including consent withdrawal, please see our Privacy Policy

Reviewer #1: No

---

## [Editor Report · Acceptance letter]

PONE-D-25-13410R1

PLOS ONE

Dear Dr. Xu,

I'm pleased to inform you that your manuscript has been deemed suitable for publication in PLOS ONE. Congratulations! Your manuscript is now being handed over to our production team.

Kind regards,

on behalf of

Professor Dr. Amitava Mukherjee

Academic Editor

PLOS ONE